# Eosinophilic Esophagitis—What Do We Know So Far?

**DOI:** 10.3390/jcm12062259

**Published:** 2023-03-14

**Authors:** Jakub Wąsik, Ewa Małecka-Wojciesko

**Affiliations:** Department of Digestive Tract Diseases, Medical University of Lodz, 90-419 Lodz, Poland

**Keywords:** eosinophilic esophagitis, esophagus, endoscopy, diagnosis, treatment, GI inflammation

## Abstract

Eosinophilic esophagitis is a Th-2 antigen-mediated disease in which there is an influx of eosinophils to all layers of the esophagus, triggering an inflammatory response. Chronic inflammatory process causes esophageal remodeling, leading to difficulties in swallowing. Food impaction, heartburn, and chest pain are other characteristic (but not pathognomonic) symptoms in adults. Although the disease has only been described since in the early 1970s, its incidence and prevalence are rapidly growing, especially in Western countries. According to the diagnostic guidelines, there should be at least 15 eosinophils visible per high-power field in biopsies obtained from different sites in the esophagus upon endoscopy with relevant esophageal symptoms. Other diseases that can cause esophageal eosinophilia should be ruled out. Eosinophilic esophagitis treatment may be challenging; however, new methods of management have recently emerged. The currently used proton pump inhibitors, topical corticosteroids, and elimination diet are combined with biological treatment. New methods for disease diagnostics and clinical course assessment are also available. This review presents current knowledge about the disease, supported by the latest research data.

## 1. Introduction

Eosinophilic esophagitis (EoE) is a chronic inflammatory disease. The first cases of EoE were described in the 1970s [1,2]. Although half a century will soon have passed since its initial description, we still have many unknowns. 

The literature describes an increasing incidence of EoE in recent decades, consisting of approximately five to ten new cases per 100,000 inhabitants annually for both children and adults [3]. This consequently contributes to the growing prevalence, which was 15.4 per 100,000 inhabitants before 2007 and has already reached 63.2 per 100,000 inhabitants since 2017 [3].

The etiopathogenesis of this disease has not yet been fully elucidated. Regarding risk factors, we still rely on individual reports. The factors that have been extensively studied in recent years are place of residence, race, gender, and age.

The current data show that the immune cells present in the esophagus, which are hyperstimulated by antigens (for example, food antigens), release numerous pro-inflammatory cytokines such as interleukin (IL)-4, IL-5, and IL-13 [4]. This drives the entire mechanism of the disease as eosinophils are recruited to stimulate a further inflammatory response [5]. As a result of chronic esophagitis, subepithelial fibrous remodeling occurs. If uncontrolled, this may adversely affect the esophagus, leading to motor disorders, dysphagia, and the formation of strictures, which may eventually lead to food bolus impaction [6].

The disease may occur at any age. Risk factors for EoE include the being male, Caucasian, and having an atopic disorder such as rhinitis or asthma [7]. Currently, EoE is the second-most prevalent cause of chronic esophagitis after gastroesophageal reflux disease (GERD) [8]. It is worth mentioning that this disease is now the most common cause of esophageal food bolus impactions in patients presenting to emergency departments [9]. EoE is associated with impaired quality of life in many domains, not necessarily due to the disease symptoms [10].

## 2. Epidemiology

It is difficult to provide accurate data on the epidemiology of EoE because the reported data vary. In a 2019 systematic review with a meta-analysis by Navarro et al., which collected data on the prevalence and incidence of EoE in the general population, the prevalence of EoE was 34.2 cases per 100,000 inhabitants [11]. In the same study, the overall incidence rate was 4.4 new cases of EoE per 100,000 inhabitants/year [11]. EoE occurs three times more often in men than in women, but there are no sex differences concerning the severity of the disease [12,13]. A study conducted by Schreiner et al. in Swiss patients showed that although the incidence is much higher in men and they had more severe endoscopic changes, their Eosinophilic Esophagitis Activity Index (EEsAI) and EoE-specific quality of life (EoE-QoL-A) were not different from those measured in women [14]. In a study in the Dutch population from 1995 to 2019, the mean age at diagnosis in adults was 42.9 ± 15.4 years, which was significantly later in women compared to men (44.5 ± 16.5 years vs. 42.2 ± 14.8 years; *p* < 0.001) [13]. 

Overall, it can be seen that there is an increasing trend of incidence and prevalence [9,11,15]. In the Netherlands, the incidence rates rose from 0.01 (95% CI: 0.0–0.04) new cases per 100,000 inhabitants in 1995 to 3.16 (95% CI: 2.90–3.44) new cases per 100,000 inhabitants in 2019 [13]. Regarding prevalence, the data are very divergent: from 2.3/100,000 in Denmark to 90.7/100,000 in Ohio [9]. However, it can be assumed that the frequency of EoE is higher in Western countries than in the East [8,9]. EoE is most common among Caucasians [15]. In a pediatric population, a study conducted by Zdanowicz et al. in northeastern Poland demonstrated that the average annual incidence rate of EoE was 2.83 cases per 100,000 children between 2013 and 2018 [16]. In some centers, the frequency may be higher, as demonstrated in a study of Jordanian children between 2015 and 2020 in which 3% of patients were diagnosed with EoE [17]. 

Although there are many descriptions of increasing prevalence and incidence in the literature, to the best of our current knowledge there was only one study on mortality in EoE. In a nationwide, population-based matched cohort study in Sweden, there were 1625 patients diagnosed with EoE over 12 years (2005–2017) [18]. In the EoE group, there were 4.6 deaths per 1000 person-years, similar to 4.57 deaths per 1000 person-years in population comparators (HR = 0.97, 95% CI = 0.67–1.40). There was also no association with higher mortality from cancer or cardiovascular diseases. Additionally, mortality rates in patients with EoE did not demonstrate a difference between sexes. These results may be reassuring, although more population studies are needed to draw further conclusions.

An interesting issue is the seasonality of the morbidity and severity of symptoms. There was no correlation between the season of the year and the number of newly diagnosed patients in the Dutch study [13]. However, according to studies conducted on children in the United States, 45% of the cases were diagnosed in the spring [19]. Moreover, in an American study on the seasonal exacerbation of EoE in adults and children, 71% of cases occurred in the fall and summer months [20]. In those patients, typical rings (92% vs. 35%; *p* = 0.001) and strictures (62% vs. 22%; *p* = 0.002) were detected more often in an upper G.I. endoscopy. Consequently, they were associated with a worse EoE Endoscopic Reference Score (EREFS), a classification system of endoscopy findings in EoE which will be discussed in more detail later in this article (1.7 vs. 3.7; *p* = 0.01) [20].

Another study from the United States, conducted in Nevada between 2013 and 2017, showed that in follow-up testing, individuals living in the northern region of the state experienced 1.95 higher odds of an EoE visit when compared to their southern counterparts (OR = 1.95, 95% CI: 1.68, 2.26, *p* < 0.001) [21]. This may be connected to the dry climate of Nevada, which has a longer pollen season and differs significantly from the mild climate of the Netherlands [21]. Among children from northeastern Poland, seasonal allergies were observed in 23.08% of children with EoE compared to 8.82% in children without EoE (*p* = 0.005). The incidences of food allergies in children with or without EoE were similar: 11.1% vs. 2.52%, respectively (*p* = 0.005) [16].

## 3. EoE Risk Factors

Risk factors for EoE can be divided into genetic, environmental, and biological factors. So far, there is no single, specific etiological factor for this pathology. Therefore, it can be assumed that this is a multifactorial disease. 

### 3.1. Genetic

As previously mentioned, one of the risk factors for EoE is being male. This may be related to thymic stromal lymphopoietin (TSLP), which is a cytokine produced by epithelial cells [22]. TSLP promotes a Th-2 immune response and is overexpressed in the suprabasal compartment of the esophageal epithelium in patients with active EoE (compared to patients without EoE or with inactive EoE) [23,24]. The TSLP receptor is encoded in a pseudo-autosomal region on Xp22.3 and Yp11.3. Single-nucleotide polymorphism (SNP) in this region was associated with the disease in male patients with EoE (*p* = 0.039; OR, 2.05) [25].

Other studies show that the CCL26 gene, which encodes the chemokine eotaxin-3 involved in the transport of eosinophils to the esophagus, is highly expressed in patients with EoE, and its expression correlates with the eosinophil count in esophageal biopsy specimens [26,27].

Another genetic risk factor is the expression of calpain 14 (CAPN14), which is a calcium-activated intracellular regulatory protease that is overexpressed in esophageal epithelial cells in EoE [28,29]. Its expression is also higher following the exposure of the esophageal epithelial progenitor cell line to IL-13 (100 ng/mL) [28]. Increased levels of IL-13 are characteristic in EoE, and the interleukin is responsible for reducing the expression of desmoglein-1, which forms desmosome (cell–cell junctions in epithelial cells) [30]. Those junctions prevent tissue injury after mechanical stress [30]. A retrospective study in children diagnosed with EoE suggests that the genetic variability of CAPN14 is associated with an earlier age of disease onset [29].

### 3.2. Biological

Regarding infectious risk factors for EoE, the exact relationship of EoE with *Helicobacter pylori*, *herpes simplex virus* (HSV), or *Mycoplasma pneumoniae* infections has not yet been established [27,31]. It was shown that the risk of EoE increased as the prevalence of *H. pylori* decreased in Western countries over the past several decades. The latter was also connected to the possibility of polarization toward Th-1 immunity, which confers protection against Th2-mediated allergic disorders such as asthma [3]. A meta-analysis of 11 observational studies comprising data on 377,795 individuals worldwide showed that *H pylori* exposure was associated with a 37% reduction in the odds of EoE (OR = 0.63, 95% CI = 0.51–0.78) [32]. A possible association of EoE with HSV and *Mycoplasma pneumoniae* has been proposed based on case reports and case series; however, as in the case of *H. pylori*, this requires further research [31]. A 2020 study by Asfari et al. [24] on a very large population of patients demonstrated that people infected with HIV have a statistically higher rate of EoE compared to uninfected people (OR 2.1, 95% CI = 1.26–3.5, *p* = 0.004) [33]. This is very clinically relevant due to the coexistence of opportunistic infections in people with HIV. Oral candidiasis and esophageal candidiasis are some of the most common oral lesions associated with HIV and, in the context of EoE, may occur as a complication of treatment with topical steroids [34,35]. PPIs, which are commonly used in EoE, increase the gastric pH and reduce the bioavailability of antiretroviral drugs, such as rilpivirine and the protease inhibitor atazanavir, by reducing their solubility [36]. Therefore, the selection of an adequate treatment should be considered when EoE is accompanied by other diseases.

A diagnosis of celiac disease also increases the risk of EoE. However, there is no evidence that EoE is associated with the human leukocyte antigen (HLA) genetic locus [24].

It was suggested that the use of antibiotics early in life may contribute to the increased incidence of EoE in adults [37,38]. The relationship between antibiotic use and the development of EoE in the pediatric population has been evaluated; however, no such studies have been performed in adults [37]. As noted by Jensen and Dellon [31], it is unknown whether antibiotics are the true causal agent or whether their effect is intermediary in some other mechanistic pathway. Other early-life EoE risk factors described in the literature related to childbirth and the first months of a child’s development are: cesarean delivery or preterm delivery and non-exclusive breastfeeding [4,39]. 

### 3.3. Environmental

There is no evidence of the relationship between EoE and stimulants, such as alcohol or cigarettes, as only a few such studies have been conducted [31,40,41]. In one study, which compared cigarette smoking and alcohol consumption in people diagnosed with and without EoE, fewer subjects with EoE ever smoked cigarettes when compared to the non-EoE controls. (23% vs. 47%, *p* < 0.001) [40]. This study also showed that there is a lower rate of EoE among current smokers compared to non-EoE controls (5% vs. 11%); however, there is a less statistical result (*p* = 0.04). In summary, the adjusted OR of EoE based on cigarette smoking was 0.47 (95% CI = 0.24–0.92). Interestingly, opposite results of this study are seen with both daily and current alcohol consumption—the adjusted OR of EoE was 1.05 (95% CI = 0.54–2.03) and 1.57 (95% CI = 0.80–3.08), respectively [40].

There are still many issues that require further research, such as the role of socio-economic status in the occurrence and complications of EoE [42].

## 4. Etiology

Although eosinophils are present in many tissues, they are not found in the esophagus under normal conditions [6].

Under the influence of stimulating factors (still not fully known; the most often-studied factors are food allergens), epithelial and dendritic cells release epithelial cytokines (IL-25, IL-33, TSLP), leading to the activation of immune cells such as invariant natural killer T (iNKT) cells, adaptive CD4^+^ effector memory T-helper 2 (Th-2) cells, and innate lymphoid type 2 cells (ILC2) [4,5]. The main is response is the T helper 2 (Th-2) immune response, during which there is a secretion of Th-2 type cytokines, such as IL-4 IL-5, IL-13, eotaxin-3, and periostin [5,8]. The increased frequency of CD4 + Th-2 cells was found in patients with EoE in peripheral blood and esophageal biopsy specimens [5]. IL-5 stimulates eosinophils to proliferate and expand from the bone marrow into the circulating blood and then to all layers of the esophagus [6]. Activated eosinophils regulate angiogenesis and endothelial activation by releasing vascular cell adhesion molecule 1 (VCAM-1) and vascular endothelial growth factor (VEGF), which are required to recruit inflammatory cells toward the esophagus [6].

There are many studies describing the role of food antigens in the pathogenesis of EoE [10]. Several observations support the idea that food plays an important role in triggering an immune response. Firstly, a very large percentage of patients achieve disease remission if a specific food that potentially causes an allergy reaction is eliminated [43]. Secondly, the re-introduction of identified food triggers leads to disease recurrence [43]. EoE also appears to be over-represented among food allergy patients undergoing oral or sublingual immunotherapy [43]. Although there is no certain evidence that aeroallergen sensitization may trigger and worsen symptoms, the potential worsening of EoE with exposure to cross-reacting inhalant allergens contained in food should be considered [44].

The esophageal epithelium is essential to the pathogenesis of EoE because the impaired function of the epithelial barrier leads to increased permeability and contact with external factors as well as the internal environment [6]. Morphologically, in changed epithelia and submucosae, we can observe basal zone hyperplasia and dilated intercellular spaces in the suprabasal layers [26]. Other histologic findings on light microscopy with hematoxylin and eosin include eosinophilic micro-abscesses that are accompanied by eosinophils of a superficial localization and eosinophil degranulation [45]. The uncontrolled transmural inflammation of the esophagus drives a progressive remodeling of the esophageal wall with fibrosis of the lamina propria, basal cell hyperplasia, and smooth muscle hypertrophy, leading to impaired esophageal function [46]. 

Atopy has been linked to EoE, with most patients having a family history of bronchial asthma, allergic rhinitis, or atopic dermatitis [6,8]. And although these are immunoglobulin E-related diseases, IgE does not play a direct role in EoE pathogenesis. Therefore, quantitative, food-specific IgE tests and skin test results are not predictive of EoE food triggers [26,47]. However, if we consider EoE in the context of allergies, it would be worth considering the relationship between this disease and the hygiene hypothesis, which is increasingly discussed as the cause of various allergic diseases [43]. This theory shows that an excessive asepticism in the environment can have an adverse effect on the microbiome of children, which is necessary for the development of the immune system [31]. Most studies focus on the intestinal microbiota, while the role of the microbiome in the esophagus is poorly understood [31]. A study in a mouse EoE model showed that supplementation with the probiotic *Lactococcus lactis* NCC 2287 reduced the eosinophil count per high power field in esophageal tissue, but only when given as a therapeutic treatment [48]. Interestingly, supplementation with another probiotic, *Bifidobacterium lactis* NCC 2818, had no significant effect on esophageal eosinophilia [48]. 

Another immunoglobulin that has not been fully studied in the context of EoE pathophysiology is IgG4. IgG4 is a Th-2 induced isotype which is considered anti-inflammatory because it fails to engage complement due to a low affinity for C1q [49]. However, it is associated with various autoimmune diseases such as bullous pemphigoid, myasthenia gravis, and IgG4-related disease, which has some similar characteristics to EoE as a chronic inflammatory and fibrosing disorder [49]. Some studies report its higher levels in esophageal tissues, and an increased total serum level of IgG4 in reaction to specific food, such as milk or wheat [4]. This analysis may have been limited by biopsy samples, which could be subject to sampling bias in a patchy disease [4]. On the other hand, other authors did not show differences in the levels of food-specific IgG4 between EoE patients and controls [4]. A study evaluating esophageal biopsy specimens in adults and children showed that patients with EoE were significantly more likely to stain positively for IgG4 than children with reflux esophagitis or controls (*p* = 0.015) [50]. In this study, there was a statistically insignificant trend toward distal specimen IgG4 staining being associated with foreign body/food impaction, which was observed in 50% of patients with distal IgG4 staining compared to 24% of patients without (*p* = 0.153). Although IgG4-positive staining was specific to EoE (100%), it had a poor sensitivity of 48% in EoE patients [50]. EoE patients had significantly higher IgG4 serum levels than GERD patients at baseline (121.0 ± 68.1 vs. 71.1 ± 49.9 mg/dL, *p* = 0.038), and those levels decreased after budesonide therapy (121.0 ± 68.1 vs. 104.2 ± 61.3 mg/dL, *p* = 0.019) [51]. Additionally, the count of IgG4-positive esophageal plasma cells was significantly reduced (29.1 ± 26.7 before vs. 0.1 ± 0.3 after; *p* < 0.001) after budesonide therapy in the stroma and not significantly reduced (7.0 ± 15.1 before vs. 0.1 ± 0.3 after) in the papillae of the lamina propria [51]. Understanding the role of IgG4 in EoE could help develop a targeted elimination diet and possibly elaborate novel treatments based on anti-IgG4 monoclonal antibodies [49].

Although the human microbiome is being increasingly considered in the context of inflammatory and autoimmune diseases and even along the gut–brain axis, its role in the pathogenesis of EoE is unknown. Patients with active EoE present changes in the esophageal microbiota and increased or decreased levels of particular species such as Haemophilus Fusobacterium, Aggregatibacter, and Actinomyces. However, studies show conflicting results for individual families, such as Streptococcaceae [52]. It may be worth focusing on Haemophilus because some studies demonstrated that their abundance was particularly high in EoE subjects, and it is associated with a range of other Th2-mediated conditions [52]. 

## 5. Symptoms

The most commonly reported symptoms in children include vomiting (16.7–59.6% in patients), abdominal pain (15.7–56.6%), dysphagia (4.8–60.9%), and food impaction (6.7–21.7%) [6]. On the other hand, in adults, the sequence is usually dysphagia (46.2–94.5%), food impaction (16.9–65.7%), and heartburn (7.7–54.5%) [6,53]. Chest pain or exercise-induced chest pain may be other symptoms in adults, who demonstrate fewer symptoms that are typical for children, such as nausea or vomiting [54]. Between periods of exacerbation, many patients remain asymptomatic [55]. In a study on the Japanese population, 18.8% of adults with EoE remained without symptoms [56]. In the case of heartburn and regurgitation, both EoE and gastroesophageal reflux disease (GERD) can be dealt with [57]. In addition, GERD may predispose patients to EoE by impeding the integrity of the esophageal mucosa [57]. As GERD and EoE may coexist, some patients may need to be treated with both PPI and anti-inflammatory drugs [58].

As previously mentioned, the symptoms do not have to concern only the digestive system. In retrospective studies, some patients experienced respiratory symptoms such as a cough, recurrent croup, hoarseness, and throat clearing [59]. The limitation of such studies is the fact that it is not known whether the symptoms coexisted or were associated with the ongoing EoE [59].

One of the serious symptoms that requires urgent endoscopic intervention is esophageal food impaction (EFI). The risk of EFI is higher in EoE patients with esophageal furrows and/or rings [42]. This may lead to esophageal perforation [42]. Although the data spread regarding EFI in EoE patients is large, with a frequency ranging between 0.5 and 42%, endoscopists should be aware that EoE may be one of the causes of EFI [60] in patients at risk, i.e., particularly Caucasian males under 50 years of age with an atopic disease. 

## 6. Diagnostics

The diagnostic criteria for EoE are quite simple. There are three steps to making an EoE diagnosis. The first includes the above-mentioned symptoms reported by the patient, which are connected with esophageal dysfunction. Second, an increased number of eosinophils must be found in the endoscopic esophageal biopsy. The third criteria is that diseases (such as gastroesophageal reflux disease, esophageal eosinophilia, or achalasia) causing esophageal eosinophilia should be ruled out.

### 6.1. Endoscopy and Histopathology

It is important to remember that although endoscopic findings may strongly suggest the disease, they are not completely sensitive or specific to the diagnosis [61]. One such finding is the presence of exudates, which are whitish plaques in the esophagus that can easily be misinterpreted as candidiasis [61]. Another finding is the presence of rings, which can be from one to several millimeters thick. It is important that the rings persist while air is being blown into the esophagus. The case of their collapse (this is called feline esophagus) is suggestive of other gastrointestinal diseases, such as GERD [61,62]. 

In order to properly group endoscopic findings, a classification system called The EoE Endoscopic Reference Score (EREFS) was introduced. It was proposed in 2012 by Hirano et al. [63] as a response to the lack of a standardized classification to describe common abnormalities identified during endoscopy. It includes five items:

Fixed rings

Grade 0: none;Grade 1: subtle, circumferential ridges;Grade 2: distinct rings that do not impair the passage of a standard diagnostic adult endoscope;Grade 3: distinct rings that do not permit the passage of a diagnostic endoscope.

Exudates

Grade 0: none;Grade 1: involving <10% of the esophageal surface area;Grade 2: involving >10% of the esophageal surface area.

Furrows

Grade 0: absent;Grade 1: present.

Edema

Grade 0: absent;Grade 1: loss of clarity of absence of vascular markings.

Stricture 

Grade 0: absent;Grade 1: present.

Some studies show that although the classification system is good for quantifying changes during endoscopy, it does not always correlate well with the clinical and histological aspect of the disease, potentially limiting its use in monitoring the effectiveness of therapy [53]. However, this all depends on the study, as another study showed that the total EREFS score is highly predictive of EoE, and the findings are highly responsive to treatment [64]. In yet another study, all inflammatory features (edema, exudates, and furrows) significantly improved after treatment; unfortunately, the fibrostenotic features did not [65]. Table 1 illustrates the differences in the EREFS results between two studies. Undoubtedly, the advantage of EREFS is the clear and easy-to-interpret scoring of individual changes.

According to the 2022 British Society of Gastroenterology (BSG) and British Society of Paediatric Gastroenterology, Hepatology and Nutrition (BSPGHAN) guidelines, in the histopathological specimen, it is necessary to find at least 15 eosinophils per high-power field on light microscopy or at least 15 eosinophils in 0.3 mm^2^ or more than 60 eosinophils in 1 mm^2^ [47]. At least six biopsies should be obtained from the lower and upper esophagus (due to the variability of the location of esophageal eosinophilia) during esophagogastroduodenoscopy. These biopsies should be fixed in 10% buffered formaldehyde and stained with hematoxylin and eosin [41]. As mentioned earlier, a differential diagnosis is important, so it is worth taking biopsy specimens from the stomach and duodenum to exclude eosinophilic gastroenteritis [57]. 

The growing awareness of eosinophilic esophagitis caused the diagnostic delay of 12.7 years (IQR 8.0–19.1) from before 2007 to drop to 0.7 years (IQR 0.2–1.3) after 2018 (*p* < 0.001) [66]. This is an important change, as an earlier diagnosis of the disease may be associated with a shorter progression of esophageal fibrosis [66]. Although standards show that at least six biopsies at two levels in the esophagus should be obtained when EoE is suspected, the percentage of European gastroenterologists following these recommendations is low [67].

### 6.2. Other Diagnostic Tools

An additional examination to detect nonspecific esophageal motility patterns is esophageal manometry [68]. Another symptom that could be diagnosed by manometry may be absent: peristalsis, with normal or increased esophagogastric junction relaxation pressure [46].

Another useful diagnostic tool may be a functional luminal imaging probe (FLIP), which achieves a three-dimensional image of the esophageal lumen by measuring the diameter, volume, and pressure changes [69]. The FLIP is placed into the esophagus either trans-orally or trans-nasally [70]. This method involves a catheter with impedance planimetry electrodes surrounded by a balloon, which is filled with a fluid of known conductivity and volume [71]. The balloon should reach at least the esophagogastric junction [70]. The balloon is then inflated to specific volumes at specific times to assess biophysical and motor responses to distention, which can be observed using the 3D display. This display provides a color–spatial reference and a concomitant number measurement of the diameter [70,71]. This makes it possible to measure the distensibility of the esophageal wall, which is reduced in both adults and children with EoE. It is also possible to measure esophageal remodeling from fibrosis [68,69]. If esophageal fibrosis occurs, there will be no increase in the cross-sectional area in the esophagus although more fluid can be pumped into the balloon [71].

Regarding imaging tests, an endoscopy may be complemented by an X-ray using barium as a contrast to assess the esophagus. A barium esophagram plays a major role in the diagnosis of dysphagia and can be helpful in diagnosing a wide variety of causes of dysphagia, one of which is EoE [72]. Esophagrams are useful tools for assessing subtle or diffuse esophageal strictures that may be overlooked during an endoscopy [72]. A characteristic finding in EoE is the stricture of the esophagus, which can progressively increase and decrease or form a circumferential, ring-shaped stricture called a “ringed esophagus” which is more common in the upper to middle esophagus at the level of the aortic arch [45]. However, esophagrams are used not only in early diagnostics but also in treatment because they may be useful in dilations planned to assess the minimal and maximal diameters of the esophagus in an attempt to quantify the fibrostenotic burden [45].

Furthermore, there is growing interest in developing non-invasive markers that are useful in EoE diagnosis.

One of these directions may be the search for miRNA-4668 in salivary and esophageal secretions. The increased expression of miRNA-4668 was found in the saliva of patients with EoE [73]. Another marker may be the spectrometric test for the presence of 3-Bromotyrosine in urine; 3-Bromotyrosine is a marker of eosinophil activation and its increased concentration was found in patients with EoE [74]. On the other hand, no association was found with fractional exhaled nitric oxide (FeNO), a marker of eosinophilic asthma [74].

### 6.3. Differential Diagnosis

An important element is carrying out a differential diagnosis. As previously mentioned, EoE should be differentiated from other pathologies with a similar clinical presentation, such as GERD. Considering a histological examination, other diseases that could cause eosinophilic esophagitis should be ruled out, including esophageal eosinophilia, achalasia, infections, connective tissue disorders, drug hypersensitivity reaction, eosinophilic gastroenteritis hypereosinophilic syndrome, and Crohn’s disease [8,58,75]. Some patients may have asymptomatic esophageal eosinophilia, which makes the diagnostic process difficult. They may also have EoE in which dysphagia will be masked by adaptive processes, demonstrating how important endoscopic examinations and manometry are in the diagnostic process of an upper G.I pathology [54]. 

## 7. Treatment

Treatment of eosinophilic esophagitis can be divided into pharmacological and non-pharmacological. Non-pharmacological treatments include diet modification and endoscopic esophageal procedures for the management of complications [34]. The main goal of treatment is to eliminate inflammation, the effect of which will be obtained in remission as a decrease of eosinophils to below 15 per 0.3 mm^2^ and a normal endoscopic image of the esophagus [41,47]. An esophageal biopsy should be repeated 8–12 weeks after the start of treatment to assess its effect (see Figure 1) [47]. The term “3D therapy” is often used in treatment to reflect the three most important aspects of treatment—drugs, diet, and dilatation [34,41]. Interestingly, a survey of adult and pediatric gastroenterologists in Europe and the United Arab Emirates showed that the awareness of current guidelines varies by country [67]. 

### 7.1. Dietary Treatment

In the 1990s, the first studies considered food allergens to be the main antigenic trigger of EoE. This led to testing the elimination of six food groups that account for the majority of IgE-mediated food reactions (milk, wheat, egg, soy, nuts, and seafood) in many pediatric and adult studies with high clinical and histologic remission rates [76]. This contributed to the creation of less restrictive diets, as it was proven that nuts and fish/seafood rarely trigger EoE, and the most common food triggers were milk and gluten [76]. Three dietary exclusion diets are considered. Diet-related clinical improvement by excluding certain food antigens suggests that food induces a CD4 + Type-2 allergic inflammatory response [5]. The most restrictive diet is a six-food elimination diet (6-FED), comprising the avoidance of cow’s milk, egg, soy, wheat, peanuts, and seafood. A less restrictive option is the four-food elimination diet (4-FED), which eliminates cow’s milk, wheat, egg, and soy. The least restrictive and the most comfortable diet for the patient is the two-food elimination diet (2-FED) which involves the removal of cow’s milk and wheat. Unfortunately, being the most comfortable for the patient does not mean that it will guarantee the greatest effectiveness. Studies have shown that for the above-mentioned diets, the highest remission was achieved when six components are eliminated in 50–71% of patients [4,77]. In the case of the diets excluding four and two components, remission was achieved in 46–54 and 43–44% of cases, respectively [10,77]. Therefore, considering the patient’s habits, a step-up empiric approach to food elimination was proposed, beginning with the two-food elimination diet (2-FED) [78]. If remission is not observed after 8–12 weeks from the treatment, 4-FED is initiated, and if this diet fails, 6-FED is introduced [78]. Theoretically, it seems that it would be most convenient for the patient and potentially make the greatest improvement to eliminate specific antigens from the food after allergy testing. However, this method is not recommended due to the low correlation between skin tests and identified EoE-triggering foods [10]. 

### 7.2. Proton Pump Inhibitors

The primary drugs used in the treatment of EoE are proton pump inhibitors (PPIs). Their role, in addition to reducing the secretion of gastric acid by inhibiting parietal cell H+-K+ATPases, is to reduce the expression of eotaxin-3, a Th-2 cytokine involved in inflammation [57]. They can also inhibit the expression and inflammatory functions of adhesion molecules, such as oxidative burst [8]. Therefore, PPIs demonstrate many non-antisecretory activities. The optimal dose in adults is 20–40 mg of omeprazole or an equivalent twice per day for 8 weeks [77]. The dose can be increased if the response is suboptimal because there is dose–response relationship [77]. Regarding EoE complications, they may contribute to inhibition of dietary protein digestion and development of IgE antibodies in response to those proteins [31].

### 7.3. Swallowed Topical Corticosteroids

Another group of drugs used in the treatment of EoE are swallowed topical corticosteroids (STCs). They are either applied orally as a nebulized liquid using a spray (fluticasone), as a powder (fluticasone), or by using a viscous preparation of liquid budesonide [79]. The role of STCs in EoE is to modulate the immune system by suppressing inflammatory response stimulators, such as eosinophilic infiltration and T-cell infiltration, reducing interleukin 13-transcription [4]. Unfortunately, the long-term effects of STCs are unknown, as are their adequate dosage and follow-up strategies [79]. The recommended dose of fluticasone is 440 to 880 μg twice daily, while the recommended dose of budesonide is 1–2 mg twice daily. Similar to PPIs, these drugs should be taken for 8 weeks, followed by an endoscopy with a biopsy to assess treatment results. [7]. An orodispersible budesonide tablet was introduced in 2018; this is the first drug for EoE treatment approved by the regulatory authorities in Europe but not in the United States [80]. Although allergies to budesonide are rare, it can have side effects in the form of delayed hypersensitivity reactions, which have been published in the literature [80]. Local complications of steroids are esophageal candidiasis, which occurs in 5–30% of patients, and oral candidiasis, which is far less common [34].

In 2018, a randomized controlled trial was conducted in the USA to compare the effectiveness of fluticasone from a multi-dose inhaler (MDI) and an oral, viscous budesonide (OVB) slurry in patients with a new diagnosis of EoE, which they took for 8 weeks twice daily [81]. There was no difference in change in the peak eosinophil count from baseline between the OVB and MDI and no difference in the change in EREFS score [81]. The results of selected randomized controlled trials associated with STCs are listed in Table 2.

When comparing the effectiveness of individual STCs, the effectiveness of PPIs and STC therapy should also be compared. A systematic review with a network meta-analysis showed that there is no statistically significant difference in histological or clinical response between these three drugs: fluticasone, budesonide, or esomeprazole [84].

### 7.4. Biological Treatment

As EoE is a chronic inflammatory disease, specific immune targets are the next frontier in EoE therapy. A biological treatment that may have multiple target points is currently being tested. Similar to asthma and atopic dermatitis, it will be useful to use this medication class in patients with EoE who are refractory or intolerant to the standard, above-mentioned treatments [85]. As more is known about the safety, efficacy, and costs, these medications could be used earlier in the algorithm of EoE treatment, especially for dupilumab, which is also used as a treatment for multiple atopic conditions [85]. 

Research is being conducted on mepolizumab and reslizumab, which both block the binding of IL-5 to its receptor, inhibiting the activation of eosinophils [86]. Mepolizumab is a fully-humanized, anti-IL-5 monoclonal antibody. In a randomized, placebo-controlled, double-blind trial, five adults with active EoE were given two 750 mg intravenous infusions of mepolizumab 1 week apart. After 8 weeks, the patients received two further doses of 1500 mg that were 4 weeks apart [87]. Six other EoE patients received matched doses of a placebo. Although the numbers of eosinophils were significantly reduced in esophageal tissue (*p* = 0.03), the small improvements in clinical symptoms (especially related to dysphagia) were statistically non-significant. Another trial was conducted with reslizumab—patients between the ages of 5 and 18 with active EoE were randomly assigned in a 1:1:1:1 ratio to receive intravenous infusions of 1, 2, or 3 mg/kg reslizumab or a placebo at weeks 0, 4, 8, and 12 [88]. Conclusions from the results of this study are similar to the previously cited study with mepolizumab. The percentage improvement from baseline in the peak esophageal eosinophil count was greater in all three reslizumab treatment groups than in the placebo group (*p* < 0.001). There were no significant differences between the reslizumab groups and the placebo group in the patient’s eosinophilic esophagitis predominant symptom assessment scores and Children’s Health Questionnaire (CHQ) scores. 

Another target is the blockage of IL-13, which, as previously mentioned, leads to epithelial barrier dysfunction. QAX576 is an antibody that inhibits the activity of T-cell-secreted IL-13 [86]. In a clinical trial, patients with active EoE were randomly assigned in a 2:1 ratio to receive either intravenous infusions of QAX576 (6 mg/kg) or placebo at weeks 0, 4, and 8 [89]. The mean esophageal eosinophil count decreased by 60% with QAX576 compared with an increase of 23% with the placebo (*p* = 0.004). There were trends for the improved frequency and severity of dysphagia in the QAX576-treated patients. 

The last interleukin worth mentioning is IL-4. Dupilumab is a human monoclonal antibody that targets the IL-4 receptor alpha chain (IL-4Rα), leading to the inhibition of IgE production, although the most important in factor EoE is Th-2 response inhibition [90]. It is worth mentioning that dupilumab was approved by the FDA in 2022 as the first drug for EoE treatment in people aged 12 years or older and weighing at least 40 kg [91]. The recommended dose is 300 mg per week [91]. The latest randomized controlled trial on dupilumab showed that the majority of patients taking this drug at a weekly dose of 300 mg had statistically significant improvements in histologic outcomes and reductions in symptoms of eosinophilic esophagitis compared to the placebo [92]. One of the indications for dupilumab in EoE, apart from atopic diseases, is a severe form of the disease, i.e., patients with clinically significant esophageal strictures and those with failure to thrive, weight loss, or growth impairment due to EoE [93].

Another possibility is omalizumab, which is a humanized monoclonal antibody that limits the activity of mast cells by binding to IgE [94]. In a randomized controlled trial, EoE patients were treated with omalizumab or a placebo subcutaneously every 2 to 4 weeks for 16 weeks, using a weight- and serum-IgE-based dosing protocol [94]. In the omalizumab trial, treated subjects had no significant reduction in esophageal eosinophil content and no decrease in symptoms relative to placebo controls. One of the newest discoveries is anti-Siglec-8 therapy, which focuses on inducing the death of eosinophils and inhibiting mast cell activity [95]. Siglec-8 is an inhibitory surface receptor expressed on eosinophils, mast cells, and basophils [95]. Preclinical in vitro studies with this antibody were conducted, as was a Phase 2 study on patients with eosinophilic gastritis and duodenitis [95]. 

### 7.5. Complications Treatment

In more severe cases, in the presence of esophageal strictures that significantly affect the patient’s functioning, the dilatation of the esophagus is considered. However, despite bringing immediate relief, the effect is only symptomatic [96]. Improvement is observed in a large percentage of patients who undergo dilatation; the meta-analysis showed that this percentage of patients was 84.95% (95% CI = 81, 72–87, 93) after the median follow-up period of 12 months [97]. Studies have shown that this is a safe method with a low percentage of esophageal perforation (<1%), and a possible side effect is chest discomfort [7,96]. If the constriction segment is short (1–2 cm), a balloon dilator is used, and if the constriction is longer, a Savary’s dilator is recommended [7].

### 7.6. Long-Term Follow-Up

Although almost 50 years have passed since the first cases of EoE were described, information on the long-term course of the disease is still scarce in the literature. In a study conducted in patients with active EoE who had been diagnosed more than 10 years ago, 50.3% reported ongoing PPI therapy. Interestingly, no current therapy for their EoE was reported by 33.3% of patients. The number of symptoms reported by patients has decreased over the years. Additionally, EoE was rarely associated with a significant decrease in quality-of-life metrics, with 57.1% and 17.9% reporting trivial to mild decreases in QoL scores, respectively [98].
Figure 1Proposal of EoE management.
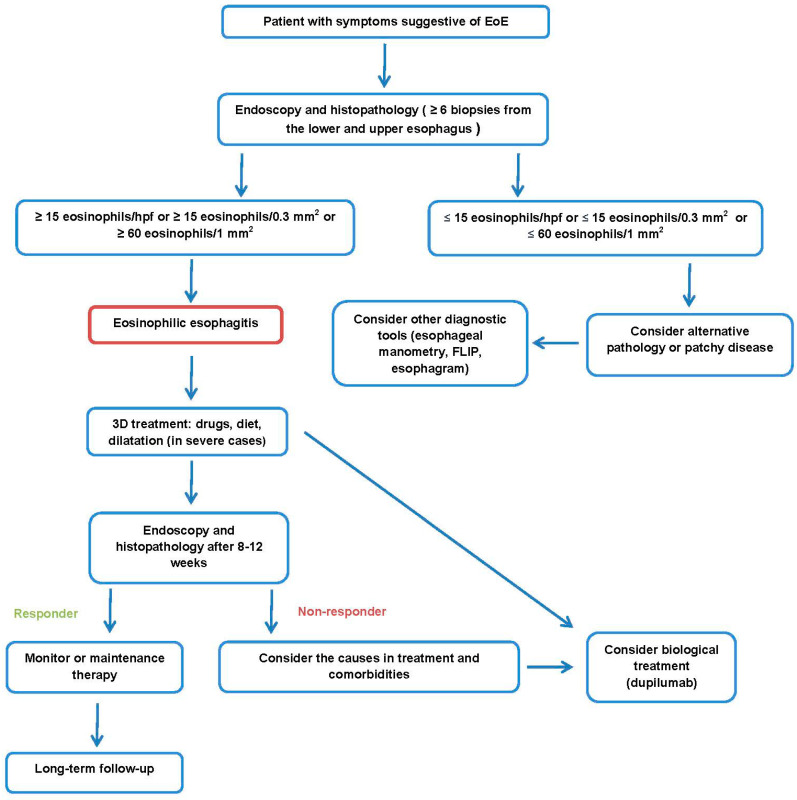


## 8. EoE Differences in Pediatric and Adolescent Patients

### 8.1. Symptoms

The symptoms of EoE vary with age. In toddlers and infants, the most frequent symptoms are feeding difficulties and failure to thrive. School-aged children present with nausea, vomiting, regurgitation, and abdominal pain. In adolescents, the symptoms are similar to those in adults, i.e., dysphagia, heartburn, food impaction, and chest pain [99]. It is also worth noting that children with EoE are more likely to suffer from atopic diseases [99].

Children, just like adults, can compensate for the symptoms of dysphagia and food retention through eating habits (excessive chewing) and dietary changes (preferring liquid meals). Therefore, it is important to conduct a thorough dietary interview with the child and the parent [100].

### 8.2. Diagnostics

In a meta-analysis of the endoscopic features of 4678 patients with EoE, rings and strictures were more prevalent in adults (57% and 25%, respectively) than in children (11% and 8%, respectively; *p* < 0.05 for each). Interestingly, white plaques and pallor or decreased vasculature were more prevalent in children (36% and 58%, respectively) than in adults (19% and 18%, respectively; *p* < 0.05 for each) [101]. For children (as for adults), the EREFs score is a useful scale for accurately identifying those with EoE. The visual detection of more than one esophageal abnormality during diagnostic endoscopy identified children with EoE with a 89.6% sensitivity and a 87.9% specificity [102]. It is also useful for treatment monitoring—in post-treatment EoE patients, the EREFS inflammatory score was 2.4 in cases of active EoE and 0.5 in cases of inactive EoE (*p* < 0.001) [102]. 

### 8.3. Treatment

The most significant differences between EoE in children and EoE in adults can be observed in terms of treatment. There are still no clear guidelines for drug doses. In pediatrics, gastroesophageal reflux presents in a very similar fashion to EoE; therefore, a PPI should be the first line of therapy to rule out GERD in this population [103]. The recommended PPI dose is 1 mg/kg of omeprazole two times per day (up to 40 mg two times per day) [100]. 

The use of STCs is more complex. The recommended dose of swallowed, inhaled fluticasone propionate is 88–440 μg twice daily (max. of 880–1760 μg). It is important that patients should be instructed to puff the medication into the mouth during breath-holding and not drink, eat, or wash their mouths for at least 30 min after swallowing [100]. Regarding the safety of budesonide, it is worth mentioning that a recent integrated safety analysis was performed on six Phase 1–3 clinical trials on adults and children. A 2.0 mg budesonide oral suspension twice daily and other doses were well tolerated in both groups. The majority of adverse events (AEs) were mild or moderate in severity—the most commonly reported AEs were infections, gastrointestinal AEs, or CNS/mood AEs [104]. Moreover, a meta-analysis assessing STCs (fluticasone propionate, oral viscous budesonide, and fluticasone) in pediatric patients with a diagnosis of EoE showed that symptomatic response was 33.6% in the corticosteroids group and 21.8% in the control group (RR 1.62; CI 0.94–2.79; *p* = 0.08). The histologic response was 49.25% in the corticosteroids group and 4.16% in the placebo group (RR 11.05 [CI 3.8–32.15]; *p* < 0.0001) [105]. In a clinical trial, patients  <150 cm and  ≥150 cm height received an induction therapy of 1 mg b.i.d and 2 mg b.i.d. OVB, respectively, for 12 weeks [106]. The OVB suspension significantly reduced the mean peak eosinophil count/HPF, clinical, and endoscopic scores (*p* < 0.01). Then patients were treated with a half-dose OVB maintenance therapy for additional 12 weeks. At the end of the follow-up period (week 36), 45% patients were still in remission, but 35% showed a significant increase in the eosinophil count [106].

The most effective but also restrictive method of food treatment is the elemental diet, which involves the removal of all sources of potentially allergenic protein from the patient’s diet through the use of an amino-acid-based formula for nutritional support [100,107]. In the case of children, compliance with it may be difficult due to the unsavory composition of the mixture [100,107].

Biological treatment may also be an opportunity to treat severe cases in children. Clinical trials are underway in people under 18 on mepolizumab, dupilumab, lirentelimab (anti-Siglec-8 therapy), or benralizumab (anti-IL-5Ra) [108]. Table 3 presents different recommendations and international consensus in the treatment of children with EoE. 

## 9. Conclusions

In summary, an increasing amount is known about eosinophilic esophagitis. Although the exact etiology of EoE is not yet known, the further approval of drugs for treatment increases the chance of therapeutic success. Regarding immunology, an attempt is being made to identify a few target points. This shows that pharmacological treatment with biological drugs, steroids, and PPIs brings satisfactory results. As can be concluded from this review, the focus should be on three areas—etiology, risk factors, and treatment. Diagnosis is not mentioned because there are many diagnostic methods, and the most important thing in diagnosis is the ability to make a differential diagnosis to exclude other diseases manifested with esophageal eosinophilia. It now seems that it is worth developing non-invasive diagnostic methods, such as identifying markers from saliva or blood. The increasing number of cases is the most worrying. As this disease has been described relatively recently, it is important to research its long-term clinical course.

## Figures and Tables

**Table 1 jcm-12-02259-t001:** Comparison of EREFS features and scores for EoE cases before and after treatment.

EREFS Features (%)	Rodríguez-Sánchez et al. (2017) [53]			Dellon et al. (2016) [64]		
	Pre-treatment (n = 67)	Post-treatment (n = 117)	*p*	Pre-treatment (n = 67)	Post-treatment (n = 67)	*p*
Fixed rings (G0) *	29.9	35.6	0.40	22	46	0.001
Fixed rings (G1)	37.3	39.3	0.78	40	43	
Fixed rings (G2)	28.4	19.7	0.17	30	11	
Fixed rings (G3)	6.3	5.1	0.80	8	0	
Exudates (G0)	56.7	62.4	0.44	51	76	0.005
Exudates (G1)	26.9	26.5	0.95	28	18	
Exudates (G2)	20.9	11.1	0.07	21	6	
Furrows (G0)	22.4	41	0.01	11	52	
Furrows (G1)	77.6	59		70	42	
Edema (G0)	14.9	25.6	0.09	37	72	<0.001
Edema (G1)	85.1	74.4		63	28	
Stricture (G0)	83.6	79.5	0.66	76	75	0.84
Stricture (G1)	16.4	20.5		24	25	

* G—grade.

**Table 2 jcm-12-02259-t002:** The results of randomized trials on swallowed topical steroids (STCs).

Study Treatment	Dose	Age, Y, Mean ± SD	Baseline EREFS ± SD	Baseline Peak Eosinophil Count, *eos/hpf* ± SD	History of Atopy (%)	Duration (Weeks)	EREFS Mean Change ± SD	% Histologic Response *	Ref.
Fluticasone propionate (n = 22)	1.5 mg twice daily	41.3 ± 12.2	4.2 ± 1.8	69.2 ± 33.3	20 (91)	12	−2.9 ± 1.92	86	[82]
Fluticasone propionate (n = 20)	3 mg twice daily	36.8 ± 9.2	3.9 ± 1.7	55.1 ± 21.3	18 (90)	12	−2.2 ± 1.84	80	[82]
Fluticasone propionate (n = 21)	1.5 mg once daily	36.8 ± 11.7	4.7 ± 1.4	56.2 ± 25.9	20 (95)	12	−2.4 ± 1.85	48	[82]
Fluticasone (n = 64)	880 mcg twice daily	39.0 ± 14.5	4.7 ± 1.9	72.5 ± 59.1	50 (78)	8	−1.9 ± 2.0	64	[81]
Budesonide (n = 65)	1 mg twice daily	36.2 ± 19.1	4.7 ± 1.8	74.1 ± 48.2	46 (71)	8	−2.6 ± 1.8	71	[81]
Budesonide (n = 51)	2 mg twice daily	22.3 ± 7.9	7.7 ± 3.5	156.3 ± 97.6		12	−3.8 ± 3.9	39	[83]

* In the first study, histologic response was defined as ≤6 eosinophils per high-power field. In second study <15 *eos*/*hpf* was assumed, while in last ≤6 *eos*/*hpf*.

**Table 3 jcm-12-02259-t003:** Comparison of EoE treatment guidelines for children.

Association	Year	First Line Therapy	PPI Dose	PPI Duration	OVB Dose	FP Dose	STC Duration	Systemic Steroids
BSG and BSPGHAN [47]	2022	PPI/diet/STC	-	8–12 weeks	Requires approval from local authorities for adolescents; for children, given in age-appropriate, viscous formulations and volume	-	12–24 weeks	Not recommended
SEGHNP [109]	2020	PPI/diet/STC	1–2 mg/kg b.i.d. (max 40 mg)	-	0.5 mg/dose (2 doses a day) <10 years	400 μg/dose (2 doses a day) <10 years	-	Consider only in emergency patients with severe dysphagia and significant weight loss or severe stenosis
					1 mg/dose (2 doses a day) >10 years	800 μg/dose (2 doses a day) >10 years		
UEG, EAACI ESPGHAN, and EUREOS [110]	2017	PPI/diet/STC	1–2 mg/kg b.i.d.	-	-	-	-	Not recommended

## Data Availability

Not applicable.

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
