# Peer review of "Eosinophilic Esophagitis—What Do We Know So Far?"

_jcm, 2023, doi:10.3390/jcm12062259_

Round 1

Reviewer 1 Report

I enjoyed this review on EoE but have a few comments, mostly minor.

line 34-36: I suspect the "cytokines" on line 34 should instead say cells or immune cells

line 95-98: this section is confusing and could be clarified

line 233-234:  this seems fairly negative given that several studies have now consistently shown that food-specific IgG4 is higher in serum of EoE patients. The referenced study appears only to be from biopsies which could be subject to sampling bias in a patchy disease.   

Line 287:  "other" diseases causing eosinophilia are described later, but it would be nice to briefly touch on some examples, perhaps in parenthesis

Table 1: please reformat this table so it is easier to read. 

line 456; I am a little murky about the duration of 8 weeks. This is a common length of time before repeating an EGD/biopsy, but typically patients need to maintain a treatment for much longer as this is a chronic disease.

line 487: "the immune system is the next potential target of treatment" reads strangely after the prior section given that steroid action presumably also is largely at the level of the immune system. Something like " Specific immunologic targets are the next frontier in EoE therapy" could be better. 

line 570 and line 63: I think there is reasonably strong data that esophageal eosinophilia/EoE has increased over time. It seems that the statement on line 63 and 570 don't reflect that the data at least favor the idea that the increase is real and not an artifact of increased awareness. 

There are several areas where proof-reading the English would be helpful for polishing the manuscript

Reviewer 2 Report

This paper represents a good point of view on EoE, offering an overview aboud current knowledge about the disease, supported by 19 the latest research data.

Nonetheless, significant improvement are needed:

- A focus on pediatric cases and guidelines, with a table that resumes different recommendations and international consensus

- A Figure with a concrete proposal of EoE management, timeline from diagnosis and treatments

eosinophilic esophagitis is rarely abbreviated as it is stated at the beginning, sometimes using the britain grammatic form
